# Spraying Zinc Sulfate to Reveal the Mechanism through the Glutathione Metabolic Pathway Regulates the Cadmium Tolerance of Seashore Paspalum (*Paspalum vaginatum* Swartz)

**DOI:** 10.3390/plants12101982

**Published:** 2023-05-15

**Authors:** Liwen Cui, Yu Chen, Jun Liu, Qiang Zhang, Lei Xu, Zhimin Yang

**Affiliations:** College of Agro-Grassland Science, Nanjing Agricultural University, Nanjing 210095, China

**Keywords:** cadmium, seashore paspalum, zinc sulfate, glutathione pathway, glutathione S-transferase

## Abstract

Cadmium (Cd) is considered to be one of the most toxic metals, causing serious harm to plants’ growth and humans’ health. Therefore, it is necessary to study simple, practical, and environmentally friendly methods to reduce its toxicity. Until now, people have applied zinc sulfate to improve the Cd tolerance of plants. However, related studies have mainly focused on physiological and biochemical aspects, with a lack of in-depth molecular mechanism research. In this study, we sprayed high (40 mM) and low (2.5 mM) concentrations of zinc sulfate on seashore paspalum (*Paspalum vaginatum* Swartz) plants under 0.5 mM Cd stress. Transcriptome sequencing and physiological indicators were used to reveal the mechanism of Cd tolerance. Compared with the control treatment, we found that zinc sulfate decreased the content of Cd^2+^ by 57.03–73.39%, and that the transfer coefficient of Cd decreased by 58.91–75.25% in different parts of plants. In addition, our results indicate that the antioxidant capacity of plants was improved, with marked increases in the glutathione content and the activity levels of glutathione reductase (GR), glutathione S-transferase (GST), and other enzymes. Transcriptome sequencing showed that the differentially expressed genes in both the 0.5 Zn and 40 Zn treatments were mainly genes encoding GST. This study suggests that genes encoding GST in the glutathione pathway may play an important role in regulating the Cd tolerance of seashore paspalum. Furthermore, the present study provides a theoretical reference for the regulation mechanism caused by zinc sulfate spraying to improve plants’ Cd tolerance.

## 1. Introduction

As one of the most toxic heavy metals in the natural environment, cadmium (Cd) pollution has been the focus of attention for decades [1]. Cd comes from various sources, and human activities accelerate the release of Cd into the surrounding environment [2,3]. As a nonessential and potentially toxic element, the sources of Cd include waste generated as a result of rapid industrialization, metal mining, and sewage irrigation [4,5], the excessive use of fertilizers and pesticides, and the mining and burning of fossil fuels [5,6,7]. It is generally believed that Cd content in soil higher than 1 mg·kg^−1^ indicates that the soil is artificially polluted [8]. According to the Chinese government’s survey on soil pollution in April 2014, Cd ranked first among eight inorganic pollutants [9]. Due to the characteristics of environmental accumulation, Cd has seriously affected the use of cultivated land [10]. Previous studies have shown that the exceedance rate of heavy metals in the cultivated soil of five major grain-producing areas in China is as high as 21.49%, while the Cd pollution rate has increased most significantly from 1.32% to 17.39% in the past 20 years [11]. In China alone, nearly one-fifth of cultivated land (approximately 20 million hectares) has been polluted by Cd, arsenic, and/or lead, resulting in approximately 12 million tons of contaminated grains and an annual economic loss of approximately CNY 20 billion [12]. Cd has a half-life of 10–35 days, cannot be degraded by microorganisms, and exists in soil for a long time. As Cd is a heavy metal pollutant that seriously affects crop production and food security, Cd pollution has attracted a lot of attention [1,13].

Cd accumulates easily in plants, impacting growth. In plants, Cd toxicity can easily lead to physiological damage, such as yellowing, metabolic disorders, a reduction in photosynthesis, the inhibition of transpiration, and a reduction in enzyme activity [14]. Cd also reduces the absorption and transportation of nutrients by plants, such as iron, manganese, zinc, potassium, magnesium, and calcium, resulting in growth retardation [15]. A high concentration of Cd can even cause plant necrosis. Cd is a nonessential nutrient in plants that is easily absorbed by plants and transported to aboveground tissues, such as seeds and fruits, and then enters the food chain via crop production [16]. In the 1990s, Cd was classified as a Class I carcinogen. Due to its high solubility and mobility, Cd easily enters the food chain, and subsequently, causes serious harm to humans’ health [17,18]. Eating agricultural products with a high Cd content may lead to a variety of human diseases, including multiple cancers [19,20], lung injuries, renal tubular diseases [21], and Itai-Itai syndrome [22].

Extensive research has been conducted on the mechanism of Cd tolerance in *Arabidopsis* plants [23,24]. Researchers have found that endo-beta mannase MAN7 can regulate Cd tolerance by regulating the cell wall binding capacity of *Arabidopsis* roots [23]. In addition, Cd tolerance can be regulated by regulating calcium signal transduction related to the plasma membrane [24]. These results provide new insights for the molecular breeding of crops for Cd tolerance.

Given the serious harm posed by Cd to human beings, it is urgent and important to develop an effective strategy to reduce the toxicity and accumulation of Cd in crops. Currently, plant growth regulators (PGRs) [25] and cultivation improvement methods [26,27] are widely used. Among them, plant growth regulators mainly include the exogenous spraying of 5-aminolevolinic acid (ALA) [28], jasmonic acid (JA) [29], melatonin (N-acetyl-5-methoxytryptamine) [30], and mineral elements (such as Zn^2+^) [31]. Cultivation measures include grafting [32,33,34], the intercropping of hyperaccumulator crops [26], and the optimizing of environmental factors, such as carbon dioxide [35] and temperature [27], as well as light quality control [36], which are effective strategies for reducing Cd toxicity and accumulation in crops. 

Seashore paspalum (*Paspalum vaginatum* Swartz) is a perennial herb of the *Paspalum* genus in Gramineae that is mainly distributed in tropical and subtropical coastal areas, with multiple resistance and adaptation strategies employed in harsh environments. In particular, it has excellent properties such as heavy metal resistance and is widely used for ecological restoration [35]. Studies have shown that the plant toxicity of Cd can be inhibited by the interactions of cations (such as Zn^2+^) during the root uptake of Cd [36]. In traditional agricultural practices, zinc sulfate (ZnSO_4_) or zinc chelate of ethylenediamine tetraacetic acid chelate (ZnEDTA) is applied to leaves and the ground [37], while foliar zinc application has been shown to be an effective method for promoting plants’ absorption of zinc [38]. A study on Triticum aestivum cv. Shield plants showed that the proportion of zinc on leaves treated with ZnSO_4_ was significantly higher than that of leaves treated with ZnEDTA [38]. To further improve the Cd resistance of seashore paspalum, we adopted the method of spraying ZnSO_4_. As a simple and easy way to improve the Cd tolerance of plants, ZnSO_4_ spraying has been widely conducted [39]. However, relevant studies only revealed the protective mechanism from a physiological perspective, and the tolerance mechanism of plants was not revealed. Therefore, we used physiological and transcriptome sequencing methods to reveal the mechanism controlling the Cd tolerance of seashore paspalum by spraying different concentrations of zinc sulfate.

## 2. Results

### 2.1. Effect of Zinc Sulfate on the Accumulation of Zn and Cd in Different Organs of Seashore Paspalum under Cadmium Stress

As shown in Figure 1a, under Cd stress, with the increase in the zinc sulfate concentration, the zinc content in different organs (the root, stem, and leaf) of seashore paspalum presented a gradually increasing trend. In general, under the same zinc sulfate concentration conditions, the content of Zn in leaves was significantly higher than that in the stems and roots. The content of Zn in the stems was significantly higher than that in the roots (except in the 0 mM ZnSO_4_ treatment). The content of Zn in different organs after the 2.5 mM ZnSO_4_ treatment was 2.08 (roots) and 13.44 (leaves) times higher than that after the 0 mM ZnSO_4_ treatment; the content of Zn in different organs after the 40 mM ZnSO_4_ treatment was 29.30 (roots) and 139.28 (leaves) times higher than that after the 0 mM ZnSO_4_ treatment.

As shown in Figure 1b, compared with the 0 mM ZnSO_4_ treatment, the 2.5 mM ZnSO_4_ and 40 mM ZnSO_4_ treatments significantly reduced the Cd content in the stems and leaves. The Cd content in the stems and leaves decreased by 70.86% and 73.39%, respectively, after the 2.5 mM ZnSO_4_ treatment. The Cd content in the stems and leaves decreased by 57.03% and 62.97%, respectively, after the 40 mM ZnSO_4_ treatment.

As shown in Figure 1c, compared with the 0 Zn treatment, the 2.5 Zn and 40 Zn treatments significantly reduced the root–stem and root–leaf Cd transfer coefficients. The Cd transfer coefficients of the root–stem and root–leaf decreased by 72.78% and 75.25%, respectively, after 2.5 mM ZnSO_4_ was sprayed. The Cd transfer coefficients of the root–stem and root–leaf decreased by 58.91% and 64.48%, respectively, after 40 mM ZnSO_4_ was sprayed.

### 2.2. Effect of Zinc Sulfate Spray on the Photosynthetic Parameters of Seashore Paspalum under Cadmium Stress

The effect of zinc sulfate spray on the photosynthetic parameters of seashore paspalum under Cd stress is shown in Table 1. It can be seen that zinc sprayed at a low or high concentration increased the net photosynthetic rate, stomatal conductance value, and transpiration rate. The three photosynthetic parameter values after the 2.5 Zn treatment were significantly higher than those after the 40 Zn and 0 Zn treatments. Compared with the 0 Zn treatment, the net photosynthetic rate, stomatal conductance value, and transpiration rate of the 2.5 Zn treatment increased by 67.92%, 104.29%, and 94.95%, respectively. After the 40 Zn treatment, the net photosynthetic rate, stomatal conductance value, and transpiration rate increased by 4.79%, 16.94%, and 14.29%, respectively.

### 2.3. Effect of Zinc Sulfate Spray on the Antioxidant Metabolism of Seashore Paspalum under Cadmium Stress

#### 2.3.1. Effect of Zinc Sulfate Spray on the Active Oxygen Content of Seashore Paspalum under Cadmium Stress

The effect of zinc sulfate spray on the active oxygen content in seashore paspalum under Cd stress is shown in Figure 2. It can be seen that, in general, after all treatments, the contents of superoxide anion free radicals and hydrogen peroxide in stems were significantly higher than those in the leaves and roots, and the contents of superoxide anion radicals and hydrogen peroxide in the leaves were significantly higher than those in the roots. The contents of superoxide anion radicals and hydrogen peroxide in different parts (roots, stems, and leaves) of 2.5 Zn- and 40 Zn-treated plants were significantly lower than that of active oxygen in 0 Zn-treated plants. However, there were differences in the extent of the reduction among different treatments. Compared with the 0 Zn treatment, the 2.5 Zn treatment reduced the superoxide anion radicals in the roots, stems, and leaves by 85.85%, 41.31%, and 41.38%, respectively, and the 40 Zn treatment reduced the superoxide anion radicals of the roots, stems, and leaves by 48.51%, 16.93%, and 25.15%, respectively. In addition, compared with the 0 Zn treatment, the 2.5 Zn treatment reduced the hydrogen peroxide content in the roots, stems, and leaves by 31.78%, 16.61%, and 23.62%, respectively, and the 40 Zn treatment reduced the hydrogen peroxide content of the roots, stems, and leaves by 20.87%, 15.78%, and 9.80%, respectively.

#### 2.3.2. Effect of Zinc Sulfate Spray on the Antioxidant Metabolism of Seashore Paspalum under Cadmium Stress

The effect of zinc sulfate spray on the antioxidant metabolism of seashore paspalum under Cd stress is shown in Table 2. It can be seen that, in general, zinc sulfate spraying can significantly increase the SOD activity in different parts (roots, stems, and leaves) of seashore paspalum. Compared with the 0 Zn treatment, the SOD activity levels in different parts (roots, stems, and leaves) after the 2.5 Zn and 40 Zn treatments were 1.11–4.08 times and 1.53–3.22 times higher, respectively. In addition, compared with the 0 Zn treatment, the 2.5 Zn and 40 Zn treatments significantly increased the activity levels of APX, GR, and GST in different parts of seashore paspalum (roots, stems, and leaves). Compared with the 0 Zn treatment, the 2.5 Zn treatment increased the APX, GR, and GST enzyme activity levels in different parts (roots, stems, and leaves) by 42.19–329.41%, 75.00–128.57%, and 188.89–725.00%, respectively; the 40 Zn treatment increased the APX, GR, and GST enzyme activity levels in different parts (roots, stems, and leaves) by 8.87–35.29%, 23.08–60.00%, and 177.78–425.00%, respectively.

In addition, compared with the 0 Zn treatment, the 2.5 Zn and 40 Zn treatments significantly increased the contents of reduced glutathione and oxidized glutathione in different parts (roots, stems, and leaves) of seashore paspalum. Compared with the 0 Zn treatment, the 2.5 Zn treatment increased the content of reduced glutathione in the roots, stems, and leaves by 42.30%, 87.30%, and 58.35%, respectively; the 40 Zn treatment increased the content of reduced glutathione in the roots, stems, and leaves by 37.93%, 68.25%, and 20.82%, respectively. Compared with the 0 Zn treatment, the 2.5 Zn treatment increased the content of oxidized glutathione in the roots, stems, and leaves by 98.06%, 36.08%, and 72.13%, respectively; the 40 Zn treatment reduced the content of oxidized glutathione in the roots, stems, and leaves by 52.20%, 16.65%, and 57.35%, respectively.

In addition, compared with the 0 Zn treatment, the 40 Zn treatment significantly reduced the contents of MDA and PRO in different parts (roots, stems, and leaves) of seashore paspalum. After the 2.5 Zn and 40 Zn treatments, the content of MDA in the roots, stems, and leaves decreased by 38.36–66.40% and 10.33–37.67%, respectively, and the PRO content decreased by 53.12–73.31% and 36.34–63.33%, respectively.

The 2.5 Zn and 40 Zn treatments significantly increased the total antioxidant capacity (T-AOC) of different parts (roots, stems, and leaves) of seashore paspalum. Compared with the 0 Zn treatment, the 2.5 Zn treatment increased the T-AOC of the roots, stems, and leaves by 188.36%, 669.32%, and 138.95%, respectively; the 40 Zn treatment increased the T-AOC of the roots, stems, and leaves by 156.50%, 202.69%, and 112.69%, respectively.

#### 2.3.3. Effect of Zinc Sulfate Spray at Different Concentrations on the Expression of Gene Families Related to Cd Absorption

Through homologous comparison, we found that 12, 7, 6, 4, 5, and 7 members were annotated in the ZRT/IRT-like protein (ZIP), heavy metal transporting ATPases (HMAs), natural resistance-associated macroscopic proteins (NRAMPs), cation exchanger (CAX), yellow strip like transporter (YSL), and metal tolerance protein (MTP) families, respectively, based on the transcriptome data. The effect of different concentrations of zinc sulfate on the expression of genes related to Cd and its chelate-related transporter family is shown in Figure 3. On the whole, the gene family members of Cd and its chelate-related transporters showed a consistent trend regardless of whether there is a low or high concentration of ZnSO_4_. In particular, the gene expression of the HMA family and NRAMP family members showed a completely consistent trend. The expression levels of ZIP, HMAs, NRAMPs, CAX, YSL, and MTP family members were upregulated in five (TRINITY_DN4422_c0_g1, TRINITY_DN7203_c0_g1, TRINITY_DN13291_c0_g1, TRINITY_DN19156_c0_g1, and TRINITY_DN1784_c0_g1), five (TRINITY_DN5985_c0_g1, TRINITY_DN5981_c2_g1, TRINITY_DN908_c0_g1, TRINITY_DN2529-_c0_g1, and TRINITY_DN1718_c0_g1), five (TRINITY_DN18452_c0_g1, TRINITY_DN4119_c0_g1, TRINITY_DN16917_c0_g1, TRINITY_DN6001_c0_g1, TRINITY_DN7944_c1_g1), two (TRINITY_DN795_c0_g1 and TRINITY_DN13955_c0_g1), three (TRINITY_DN1765_c0_g1, TRINITY_DN11409_c0_g1, and TRINITY_DN1765_c1_g1), and three (TRINITY_DN1421_c0_g1, TRINITY_DN12047_c0_g1, and TRINITY_DN3072_c0_g1) members, respectively. The expression levels of ZIP, HMAs, NRAMPs, YSL, and MTP family members were downregulated in four (TRINITY_DN6170_c0_g1, TRINITY_DN565_c0_g1, TRINITY_DN1646_c2_g1, and TRINITY_DN25501_c0_g1), two (TRINITY_DN9839_c0_g1 and TRINITY_DN10530_c0_g1), one (TRINITY_DN52708_c0_g2), one (TRINITY_DN20917_c0_g1), and one (TRINITY_DN17501_c0_g1) members, respectively. In addition, one member of the ZIP, CAX, YSL, and MTP families (TRINITY_DN43139_c0_g1, TRINITY_DN1080_c0_g1, TRINITY_DN4772_c0_g1, and TRINITY_DN5933_c0_g1, respectively) was upregulated in the low-concentration zinc sulfate treatment and downregulated in the high-concentration treatment. Furthermore, the ZIP, CAX, YSL, and MTP family members each had one member (TRINITY_DN460-03_c0_g1, TRINITY_DN2329_c0_g1, TRINITY_DN4772_c0_g, and TRINITY_ DN8166_ c0_ G1, respectively) whose expression was downregulated after the application of a low concentration of zinc sulfate and upregulated after the application of a high concentration of zinc sulfate. In addition, one member of the ZIP (TRINITY_DN8263_c0_g1) and one member of the MTP (TRINITY_DN5933_c0_g1) families had no change in expression after the application of a low concentration of zinc sulfate, but exhibited a decreased expression after the application of a high concentration of zinc sulfate.

### 2.4. Transcriptome Analysis of the Mechanism of Improving the Cadmium Tolerance of Seashore Paspalum by Zinc Sulfate Spray

#### 2.4.1. Statistics Showing the Number of Differentially Expressed Genes

Statistics showing the number of genes in the differential expression gene set are shown in Table 3. It can be seen that between 2.5 Zn and 0 Zn, there were 7291 differentially expressed genes, of which 3849 were upregulated and 3442 were downregulated. There were 7781 differentially expressed genes between 40 Zn and 0 Zn, of which 3137 were upregulated and 4644 were downregulated. A Venn diagram of each group of differentially expressed genes was drawn, as required (Figure 4). In addition, the differentially expressed volcano plots between 2.5 Zn and 0 Zn and 40 Zn and 0 Zn are shown in Figure 5. Among them, there were 3343 differentially expressed genes specific to 0.5 Zn vs. 0 Zn, 3863 differentially expressed genes specific to 40 Zn vs. 0 Zn, and 3918 differentially expressed genes specific to 2.5 Zn vs. 0 Zn and 40 Zn vs. 0 Zn.

#### 2.4.2. Enrichment Analysis of the KEGG Pathway of Differentially Expressed Genes

The enrichment analysis results of the KEGG pathway of differentially expressed genes are shown in Figure 6, which shows the first 20 pathways with the least significant Q value. It can be seen from Figure 6 that between 2.5 Zn and 0 Zn and 40 Zn and 0 Zn, differentially expressed genes were enriched in drug metabolism—cytochrome P450; drug metabolism—other enzymes; galactose metabolism; glutathione metabolism; metabolism of xenobiotics by cytochrome P450; ovarian steroidogenesis; pentose and glucuronate interconversions; porphyrin and chlorophyll metabolism; retinol metabolism; starch and sucrose metabolism; steroid hormone biosynthesis; tryptophan metabolism; tyrosine metabolism; vitamin B6 metabolism. In addition, the differentially expressed genes between 2.5 Zn and 0 Zn were also enriched in amino sugar and nucleotide sugar metabolism, biosynthesis of amino acids, carbon metabolism, and glycolysis/gluconeogenesis. Furthermore, the differentially expressed genes between 2.5 Zn and 0 Zn were also enriched in the degradation of aromatic compounds, fatty acid degradation, the neurotrophic signaling pathway, and the prolactin signaling pathway.

#### 2.4.3. Zinc Sulfate Sprayed at Different Concentrations Reveals the Mechanism through Which the Glutathione Pathway Regulates the Cadmium Tolerance of Seashore Paspalum

The glutathione pathway refers to a study by Schisler et al. (2015). It can be seen from Figure 7 that zinc sulfate spraying regulated the expression of most glutathione pathway genes. Compared with the 0 Zn treatment, the genes upregulated by the 0.5 Zn and 40 Zn treatments include TRINITY_ DN7555_ c0_ g2 (putative glutathione-specific gamma-glutamyl cyclotransferase 2), TRINITY_ DN5912_ c0_ G1 (5-oxoprolinase), TRINITY_ DN2694_ c0_ G1 (glutamate cysteine ligase A and chloroplastic), and other 36 genes. Furthermore, genes downregulated by the 0.5 Zn and 40 Zn treatments include TRINITY_ DN10266_ c0_ g1 (glutathione transferase GST 23), TRINITY_ DN10477_ c0_ g1 (probably glutathione S-transferase BZ2), TRINITY_ DN11408_ c0_ g1 (probably glutathione S-transferase GSTU1), TRINITY_ DN1560_ c0_ G1 (ribonucleate diphosphate reduce small chain), and TRINITY_ DN176_ c0_ G2 (probably L-APX 8 and chloroplastic), and 24 other genes. The genes downregulated by the 0.5 Zn treatment and upregulated by the 40 Zn treatment include TRINITY_ DN6069_ c0_ G1 (glutamate reductase, chloroplastic) and TRINITY_ DN331_ c1_ G1 (putative L-APX 6). The genes up- and downregulated by the 0.5 Zn treatment include TRINITY_ DN4578_ c0_ g1 (6-phosphogluconate dehydrogenase, decarboxylating 1), TRINITY_ DN7233_ c0_ g1 (6-phosphogluconate dehydrogenase, decarboxylating 2, chloroplastic), TRINITY_ DN7637_ c0_ G1 (glucose-6-phosphate 1-dehydrogenase, cystotropic isoform), and nine other genes. However, under high and low zinc sulfate concentration treatments, the expression of these genes was different from that of the control.

### 2.5. Mechanism of Cadmium Tolerance of Seashore Paspalum by Zinc Sulfate Spray

The mechanism controlling the Cd tolerance of seashore paspalum as a result of zinc sulfate spraying is shown in Figure 8. It can be seen that a high concentration (40 mM) and a low concentration (2.5 mM) of zinc sulfate regulated the Cd tolerance mechanism via almost the same mechanism. Zinc sulfate spray increased the zinc content in the leaves and stems and reduced the Cd content and Cd transfer coefficients. Further analysis showed that zinc sulfate spraying regulated the Cd tolerance of seashore paspalum via the glutathione pathway. In the glutathione pathway, the expression of glutathione S-transferase-related genes increased the activity of GST, thus increasing the content of GSH; reducing the contents of MDA, PRO, H_2_O_2_, and OH•; enhancing the total antioxidant capacity. The plants exhibited a strong photosynthetic capacity, thus enhancing their Cd tolerance.

## 3. Discussion

As a micronutrient, zinc participates in various physiological functions of plants, and its unbalanced supply will reduce the yield [40]. Zinc is also the main component of plant protein production and ribosome development. It influences pollen tube formation, thus contributing to pollination [41,42]. Plant yield is related to photosynthesis and the chlorophyll content [43]. With the increase in the zinc dosage, photosynthesis and the chlorophyll content decrease, while with a decrease in the zinc concentration, the chlorophyll content increases [44]. Zinc also plays a role in plant hormone metabolisms, such as the biosynthesis of auxin, tryptophan [45,46], indoleacetic acid (IAA), gibberellin, nitrogen metabolism and absorption, chlorophyll synthesis and photosynthesis, and resistance to biotic and abiotic stresses [47].

Cd is a toxic trace element and one of the most toxic heavy metals in the environment; its high mobility causes serious damage to humans’ health and environmental sustainability [48,49,50]. The phytotoxicity of Cd can be inhibited via the interaction of cations (such as Fe^3+^, Zn^2+^, and Mn^2+^) during Cd absorption by the roots [51,52]. The absorption of Cd by plants’ roots is reduced due to external Zn^2+^ [53]. There are many studies on the Cd tolerance of plants regulated by Zn^2+^ [48,52,54,55], but the specific molecular mechanism has not been reported

### 3.1. Effects of Zinc Sulfate Spraying at Different Concentrations on Cadmium and Chelate-and Transporter-Related Gene Family Members in Seashore Paspalum

Essential cations, such as Zn^2+^, have a protective effect on the toxicity of Cd^2+^ [56], which is interpreted as a result of competition. Cd^2+^ and other nonessential metal ions will enter plant cells through the absorption system of essential cations [57]. We found that compared with the 0 Zn treatment, the 2.5 Zn and 40 Zn treatments significantly reduced the Cd content in seashore paspalum plants (stems and leaves). However, the reduction range was different; that caused by the 2.5 Zn treatment was significantly higher than that caused by the 40 Zn treatment in the lower stem (70.86% vs. 57.03%) and leaf (73.39% vs. 62.97%). In addition, we found that the 2.5 Zn and 40 Zn treatments significantly reduced the Cd transfer coefficient in the root–stem and root–leaf. The reduction range of the 2.5 Zn treatment was significantly higher than the that of 40 Zn treatment in the root–stem (72.78% vs. 58.91%) and root–leaf (75.25% vs. 64.48%). From the perspective of inhibiting the Cd content in plants, we suggest spraying 0.5 Zn to improve the Cd tolerance of seashore paspalum. 

Our study confirmed that zinc sulfate spraying can alleviate Cd stress and provide Cd tolerance to plants. However, some studies pointed out that zinc-dependent and zinc-binding molecules are good candidates as toxic targets of Cd^2+^. The chemical similarity of Cd^2+^ and Zn^2+^ ions makes it possible for Cd^2+^ ions to replace Zn^2+^, thus interfering with many processes that depend on Zn [58]. The indirect evidence for this effect is the presumed transcriptional upregulation of the Zn^2+^ uptake system [59], which was also observed in fission yeast [60]. This indicates that the Zn^2+^-sensing molecule may be occupied by Cd^2+^ due to Cd^2+^ exposure. These studies show that the interaction mechanism of zinc and Cd among different species is relatively complex, which means further study is needed.

There are three processes of Cd transport in plants: root absorption, long-distance transport to the aboveground parts, and storage in leaves. Cd and its chelate-related transporters mainly include zinc/iron transporters (ZRT/IRT-like protein, ZIP) [61], natural resistance-associated macrophage proteins (NRAMPs) [62], HMAs [63], metal tolerance proteins (MTP), a cation exchanger (CAX) [64], an ATP binding cassette transporter (ABC) transporter [65], and a yellow stripe-like transporter (YSL) [66]. By analyzing the expression of genes related to Cd absorption and transport, we found that the expression level of most members of ZIP, HMAs, NRAMPs, CAX, YSL, and MTP families was upregulated. Combined with the content of Zn^2+^ and Cd^2+^ in plants, we speculate that zinc sulfate spray mainly increases the expression of genes related to Cd and its chelate-related transporters, thereby increasing zinc absorption and inhibiting Cd absorption to some extent. These results indicated that these members play an important role in reducing the Cd transport coefficient and improving the Cd tolerance of plants via sulfuric acid spraying. Because there are many upregulated genes in these gene families, which members play a major role needs to be further studied.

### 3.2. The Cadmium Tolerance of Seashore Paspalum by Zinc Sulfate Spraying via the Glutathione Metabolic Pathway

Currently, the mechanism of the Cd tolerance of plants has been revealed through transcriptomics [38,67] and metabonomics [67]. Via transcriptome sequencing, we found that compared with the 0 Zn treatment, among the thirty-six genes upregulated by 0.5 Zn and 40 Zn, nineteen genes were glutathione S-transferase-related genes, and two genes were glutathione synthase-related genes; eight of the twenty-four genes downregulated by 0.5 Zn and 40 Zn were glutathione S-transferase-related genes. In addition, we found that six of the nine genes up- and downregulated by 0.5 Zn and 40 Zn were glutathione S-transferase-related genes. These genes include TRINITY_DN1348_c0_g1 (glutathione S-transferase T1), TRINITY_DN18317_c0_g1 (probable glutathione S-transferase GSTU6), TRINITY_DN11771_c0_g2 (glutathione S-transferase F12), TRINITY_DN36059_c0_g1 (glutathione S-transferase F11), TRINITY_DN20127_c0_g2 (glutathione S-transferase F10), and TRINITY_DN11771_c0_g1 (glutathione S-transferase APIC). These results suggest that glutathione S-transferase-related genes may play an important role in regulating the Cd tolerance of seashore paspalum. Other studies, such as those on melon root [67], tall fescue [38], and the mechanism of the Cd tolerance of cauliflower seedlings [38], also showed that related genes encoding glutathione S-transferase (GST) play an important role in improving the Cd tolerance of plants. However, clarifying how glutathione S-transferase-related genes regulate Cd tolerance in plants requires further research.

The activities of key enzymes in glutathione metabolism are regulated by genes related to the glutathione pathway. We found that the 2.5 Zn and 40 Zn treatments increased the GR and GST enzyme activities of different parts (roots, stems, and leaves) by 23.08–128.47% and 177.78–725.00%, respectively, which were compared with that of the 0 Zn treatment. In particular, the activity of GST increased significantly. GR catalyzes the reduction of GSH, which is a molecule involved in many metabolic regulations and antioxidant processes in plants. GR catalyzes the NADPH-dependent reaction of the glutathione disulfide bond; so, it is very important to maintain the glutathione pool [68]. GR is involved in the defense against oxidative stress, while GSH plays an important role in the cell system, including participating in the ASH glutathione cycle and maintaining thiol (-SH) and GST substrates [68]. GR and GSH play a crucial role in determining the tolerance of plants under various stresses. Some studies show that GR plays an important role in the regeneration of GSH; so, it also prevents oxidative stress by maintaining ASH [69]. It is well known that plant GST plays a role in hydrogen peroxide detoxification, cell apoptosis regulation, and plants’ response to biotic and abiotic stresses [70]. Our research showed that the activity of GR and GST increased significantly after zinc sulfate spraying. Based on the above research results, zinc sulfate spraying can play an important role in alleviating Cd stress by increasing the activity levels of GR, GST, and other enzymes, and the key genes in its pathway are worth further exploration.

The glutathione content is regulated by GR, GST, and other enzymes [69]. In particular, the contents of reduced GSH and GSSG increased significantly. Compared with the 0 Zn treatment, the 2.5 Zn and 40 Zn treatments increased the content of reduced glutathione by 42.30–87.30% and 20.82–68.25%, respectively. Tripeptide glutathione (γ Glu-cys-gly; GSH) is one of the most important metabolites in plants and is considered to be the most important defense mechanism against reactive oxygen species (ROS)-induced oxidative damage in cells. The balance between GSH and GSSG is the core component to maintaining the redox state of cells [71]. GSH is necessary to maintain the normal reduction state of cells to offset the inhibition of ROS-induced oxidative stress. GSH is a potential scavenger of ^1^O_2_, H_2_O_2_ [72,73] and the most dangerous ROS, such as OH· [74]. Our results showed that zinc sulfate spraying plays an important role in the process of relieving Cd stress by increasing the glutathione content. The exogenous spraying of strigolactones alleviated the response of melon roots to Cd stress [67], and the key factor of NO regulating the Cd stress adaptation of tall fescue [38] revealed that GSH metabolism plays an important role in regulating the Cd tolerance of plants. These results suggest that GSH metabolism may play an important role in plants under heavy metal stress [75], which is a conservative defense mechanism [76].

## 4. Materials and Methods

### 4.1. Plant Materials and Experimental Treatments

Seashore paspalum (*Paspalum vaginatum* Swartz) plants with the same growth trend were selected and transferred to 96-well plates containing ½-strength Hoagland’s nutrient solution. According to the methods of Chen et al. (2021) [77] and Ma et al. (2001) [78], 1 mM NaOH was used to keep the pH value at 5.5, and the nutrient solution was replaced every 2 days. In the Cd treatment experiment, seedlings growing for 15 days were transferred to a 0.5 μM CdCl_2_ solution. We sprayed 2.5 mM (2.5 Zn) and 40 mM (40 Zn) ZnSO_4_ solution from 4:00–4:30 every afternoon until the leaves dripped. Plants grown without zinc sulfate (0 mM ZnSO_4_; 0 Zn) were used as the control. After 7 days of treatment, seashore paspalum plants grown under different zinc sulfate treatments (0 mM, 2.5 mM, and 40 mM) were divided into three parts: roots, stems, and leaves, for sampling.

### 4.2. Determination of the Cd Content in Plants

We slightly modified the method of Chen et al. (2021) [77]: First, the roots, stems, and leaves were washed with 20 mM EDTA-Na_2_ solution for 30 min, and then with deionized water three to five times. The washed samples were dried at 105 °C for 25 min, and then at 80 °C until they were completely dry. Dried plant samples were digested with 100% HNO_3_. Then, the digest was measured using an inductively coupled plasma optical emission spectrometer (ICP-OES, iCAP 6300, Waltham, MA, USA).

The transfer coefficient (TC) was calculated as the ratio of the metal concentration in the stem or leaf to that in the root, which evaluates the ability of the plant to transport Cd from the root to the aerial part. Refer to the calculation formula of Cao et al. (2019) [79]: TC = A_Cd_/R_Cd_. A_Cd_ is the Cd concentration in stems or leaves of seashore paspalum plants; R_Cd_ is the concentration of Cd in roots.

### 4.3. Determination of Plant Photosynthetic Capacity

After 7 days of treatment, we measured the photosynthetic characteristics of plants using an infrared gas analyzer (LI-COR 6400, LICOR GmbH, Lincoln, NE, USA). Among them, the net photosynthetic rate, stomatal conductance, and transpiration rate are variables of interest. The measurement was conducted at 10:30 am, Beijing time. Three measurements were taken.

### 4.4. Determination of the Active Oxygen Content

The method described by Ma et al. (2016) [80] was slightly modified to determine the content of superoxide anion free radicals (O_2_^−·^). The determination of the hydrogen peroxide (H_2_O_2_) content was performed according to the methods described by Jiang and Zhang (2001) [81] and Ma et al. (2016) [80].

### 4.5. Determination of the MDA Content and Proline Content

A malondialdehyde (MDA) assay kit (thiobarbituric acid method) and a proline assay kit (colorimetric method) (Nanjing Jiancheng Bioengineering Research Institute, Nanjing, China) were used to determine the contents of MDA and proline (Pro), respectively.

### 4.6. Determination of the Total Antioxidant Capacity

A total antioxidant capacity assay kit (Nanjing Jiancheng Bioengineering Research Institute, Nanjing, China) was used to determine the total antioxidant capacity.

### 4.7. Determination of the Antioxidant Enzyme Activity

A total superoxide dismutase (T-SOD) test kit (hydroxylamine method), a glutathione S-transferase (GST) test kit (colorimetric method), a glutathione reductase (GR) test kit (colorimetric method), and an ascorbate peroxidase (APX) test kit (colorimetric method) (Nanjing Jiancheng Bioengineering Research Institute, Nanjing, China) were used to determine the activity levels of SOD, GST, GR, and APX, respectively.

### 4.8. Determination of the Total Glutathione (T-GSH)/Oxidized Glutathione (GSSG) Content

The contents of T-GSH and GSSG were determined using a total glutathione (T-GSH)/oxidized glutathione (GSSG) test kit (spectrophotometry) (Nanjing Jiancheng Bioengineering Research Institute; Nanjing, China) following the manufacturer’s instructions. The reduced glutathione (GSH) content was determined according to the following formula: GSH content = T-GSH content − 2 × GSSG content.

### 4.9. RNA Extraction and Transcriptome Sequencing

According to the manufacturer’s instructions (Sigma-Aldrich, St. Louis, MO, USA), RNA was extracted from the aerial parts of seashore paspalum plants in different treatments using TRIzol reagent. As described by Chen et al. (2015) [82], We checked the purity and quality of total RNA. High-quality total RNA samples were reverse transcribed into cDNA and used for the construction of a cDNA library. Sequencing was performed on Novogene’s Illumina Hiseq platform. Gene function annotation, differential expression analysis, and GO and KEGG pathway enrichment analyses were performed with the sequence information [83,84].

### 4.10. Statistical Analysis

The data processing system SPSS version 16.0 (IBM, Armonk, NY, USA) and Microsoft Excel 2012 (Redmond, WA, USA) was used to analyze the experimental results, and Scheffe’s test (*p* < 0.05) was used for analysis of variance (ANOVA).

## 5. Conclusions

Zinc sulfate spraying can improve the antioxidant capacity of plants to a certain extent and can increase the content of reduced glutathione. Glutathione S-transferase-related genes may be the key genes that regulate the glutathione content in the glutathione pathway, which plays an important role in improving the Cd tolerance of seashore paspalum. Our work provides a theoretical reference for zinc sulfate spraying to alleviate Cd stress in seashore paspalum (*Paspalum vaginatum* Swartz).

## Figures and Tables

**Figure 1 plants-12-01982-f001:**
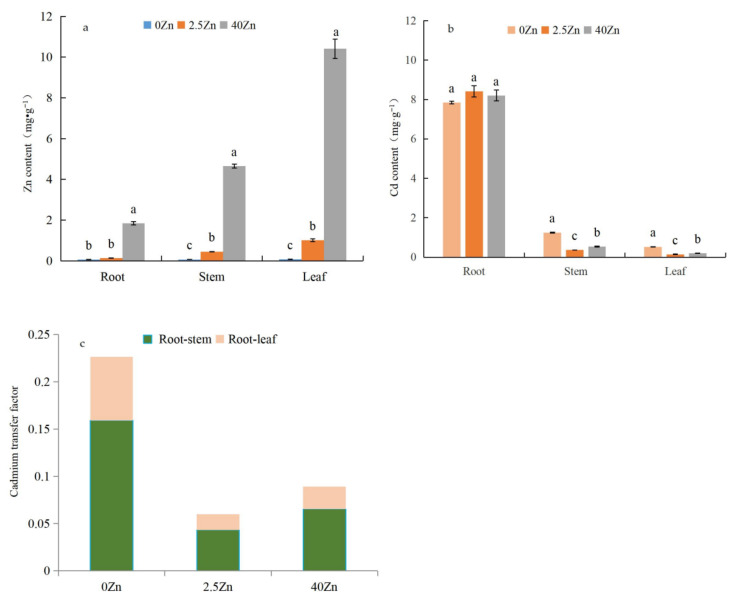
Heavy metal accumulation in seashore paspalum treated with Zn (0, 2.5, and 40 µM) under Cd^2+^ (5 µM) stress. Note: Different lowercase letters represent significant differences within columns at the 0.05 level based on Scheffe’s test. 0 Zn: 0 mM ZnSO_4_; 2.5 Zn: 2.5 mM ZnSO_4_; 40 Zn: 40 mM ZnSO_4_ (the same below).

**Figure 2 plants-12-01982-f002:**
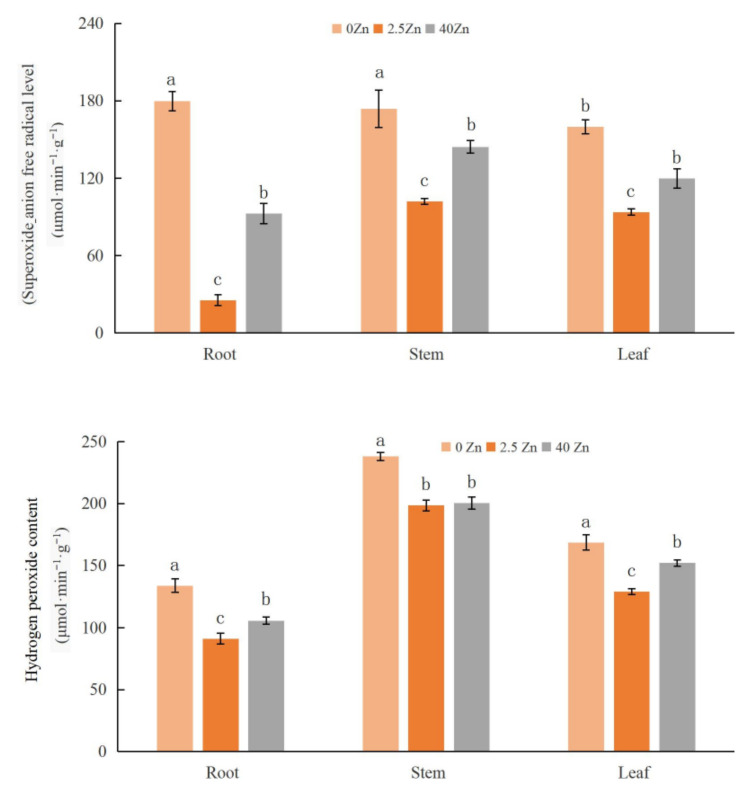
Effect of zinc sulfate spraying on reactive oxygen contents of seashore paspalum under Cd stress. Note: Different lowercase letters represent significant differences within columns at the 0.05 level based on Scheffe’s test.

**Figure 3 plants-12-01982-f003:**
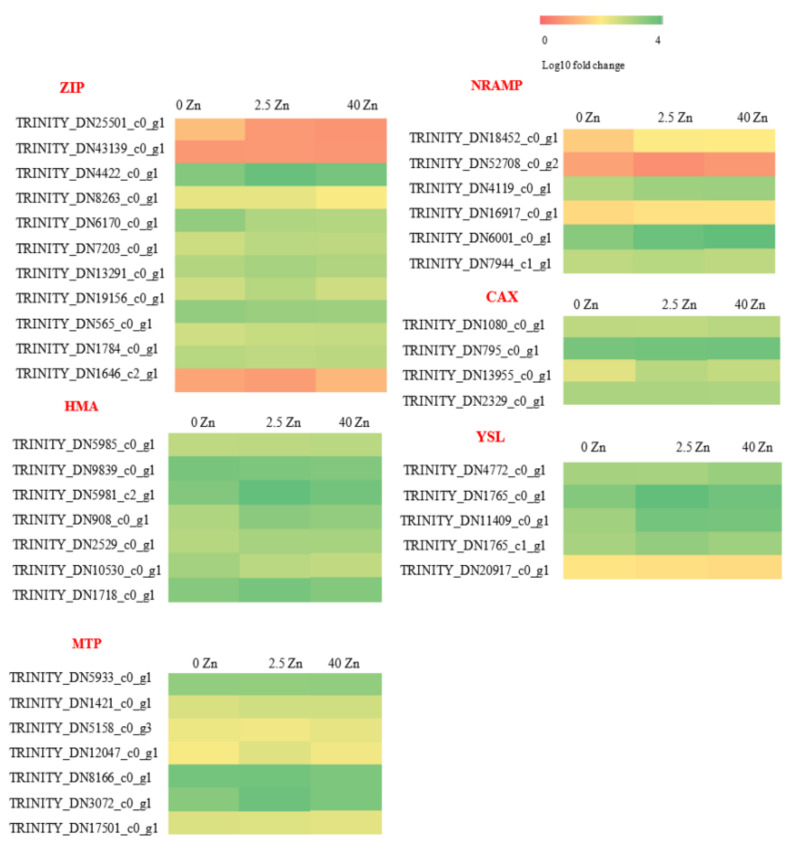
Effects of zinc sulfate spray on gene expression of cadmium uptake-related gene family members in seashore paspalum under cadmium stress.

**Figure 4 plants-12-01982-f004:**
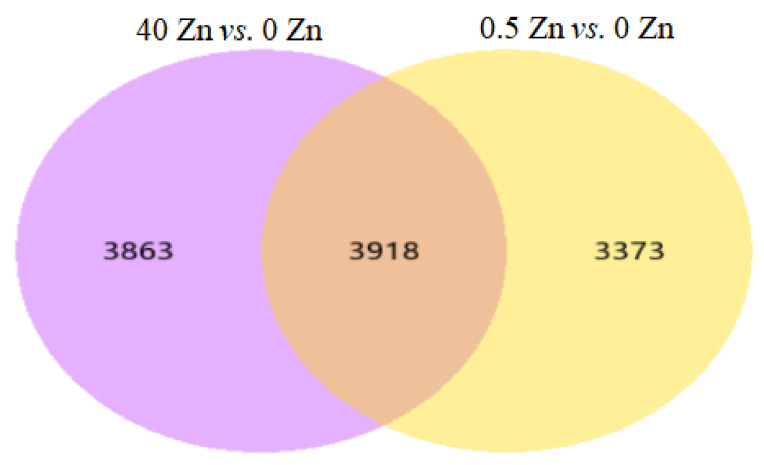
Venn diagram of differentially expressed genes.

**Figure 5 plants-12-01982-f005:**
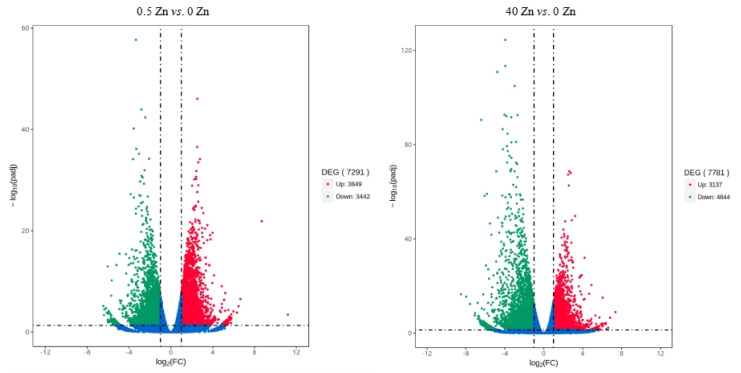
Volcano plot of differentially expressed genes. Note: (**A**) volcano plot of differentially expressed genes between 2.5 Zn and 0 Zn; (**B**) volcano plot of differentially expressed genes between 40 Zn and 0 Zn.

**Figure 6 plants-12-01982-f006:**
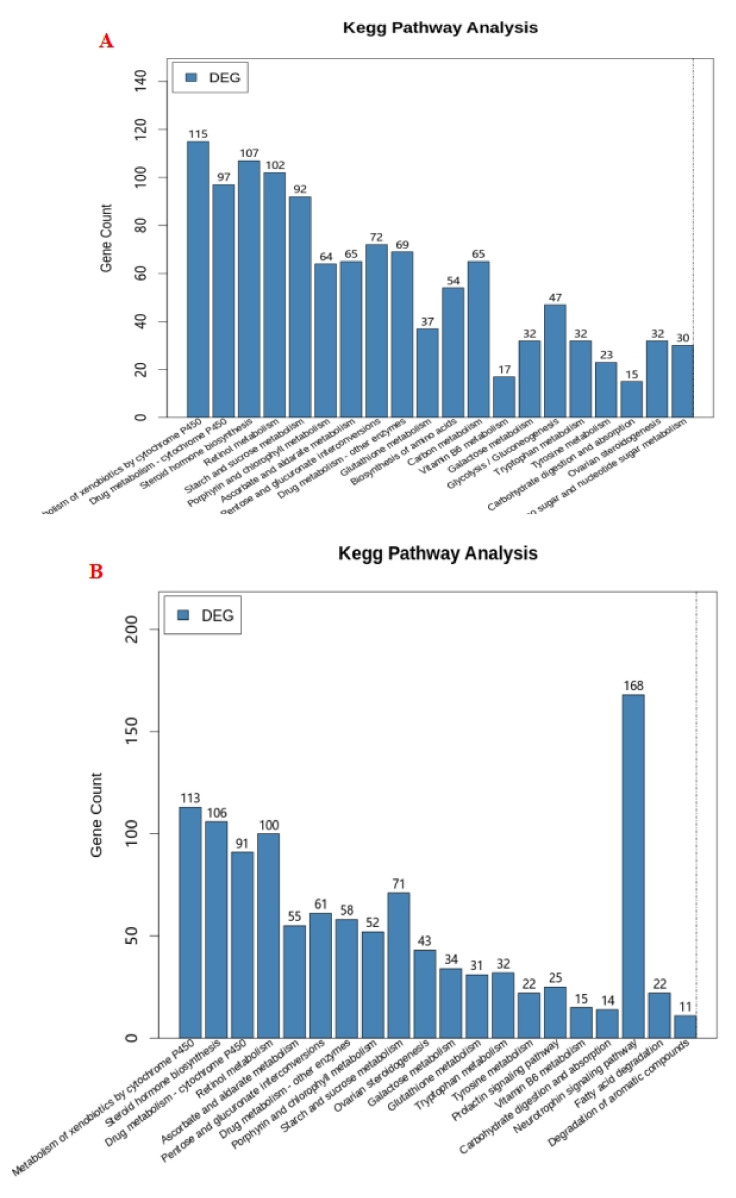
Diagram of KEGG pathway enrichment for differentially expressed genes. Note: Scatter diagram of KEGG pathway enrichment for differentially expressed genes: (**A**) between 2.5 Zn and 0 Zn; (**B**) between 40 Zn and 0 Zn.

**Figure 7 plants-12-01982-f007:**
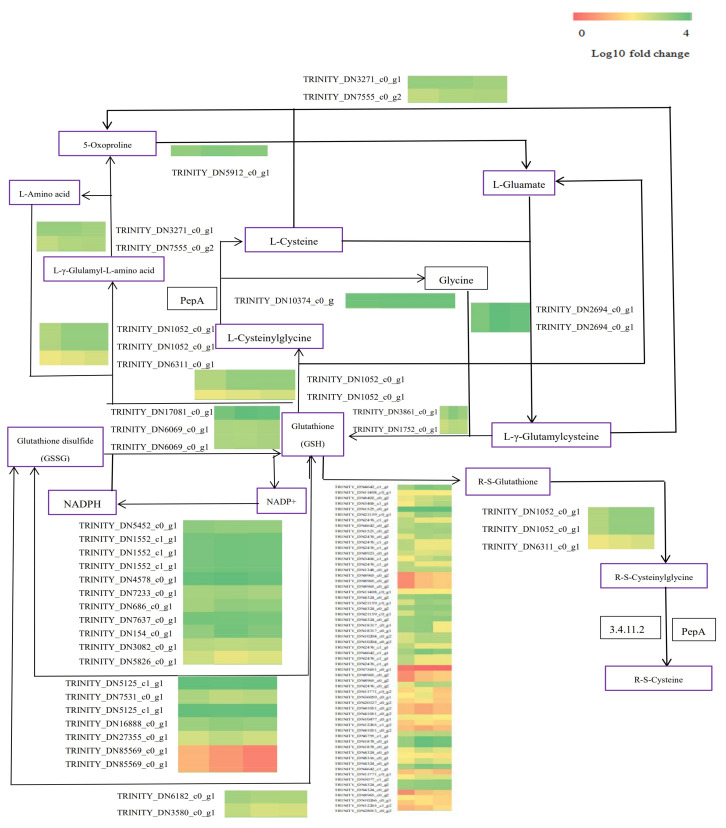
Mechanism of cadmium tolerance of seashore paspalum as a result of zinc sulfate spraying via the glutathione metabolic pathway.

**Figure 8 plants-12-01982-f008:**
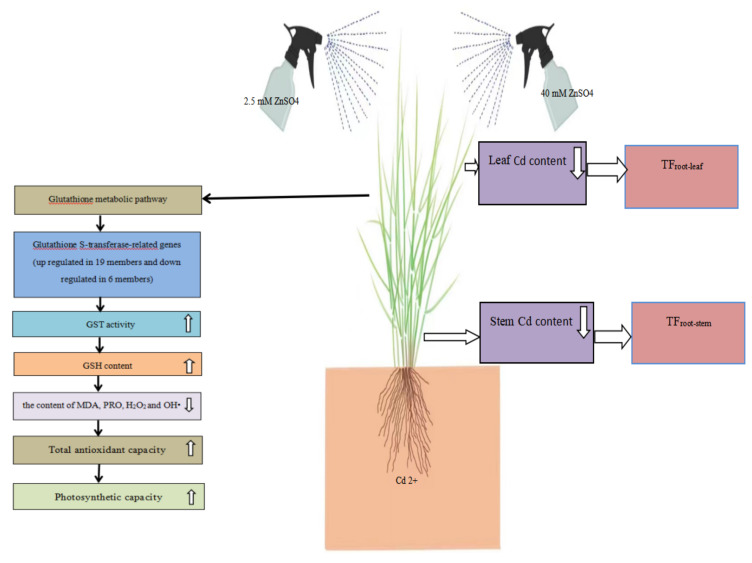
Mechanism of controlling the cadmium tolerance of seashore paspalum as a result of zinc sulfate spraying. Note: arrows indicate an increase or a decrease in the content of a substance or enzyme activity.

**Table 1 plants-12-01982-t001:** Effect of zinc sulfate on the photosynthetic parameters of seashore paspalum under cadmium stress.

Treatment	Net Photosynthetic Rate μmol CO_2_ m^−2^·s^−1^	Stomatal Conductance mol H_2_O m^−2^·s^−1^	Transpiration Rate mmol H_2_O m^−2^·s^−1^
0 Zn	1.3536 ± 0.0367 ^b^	0.0236 ± 0.0011 ^b^	0.3204 ± 0.0099 ^c^
2.5 Zn	4.2197 ± 0.0646 ^a^	0.0482 ± 0.0008 ^a^	0.6245 ± 0.0109 ^a^
40 Zn	1.4184 ± 0.0510 ^b^	0.0276 ± 0.0009 ^b^	0.3661 ± 0.0091 ^b^

Note: Different lowercase letters represent significant differences within columns at the 0.05 level based on Scheffe’s test.

**Table 2 plants-12-01982-t002:** Effect of zinc sulfate spray on the antioxidant metabolism of seashore paspalum under cadmium stress.

Parameter	Part	0 Zn	2.5 Zn	40 Zn
SOD (U/g FW)	Root	47.44 ± 4.16 ^c^	193.44 ± 4.76 ^a^	152.82 ± 7.01 ^b^
	Stem	46.16 ± 3.36 ^b^	71.35 ± 1.59 ^a^	83.98 ± 8.34 ^a^
	Leaf	50.93 ± 3.72 ^b^	78.15 ± 6.67 ^a^	56.30 ± 3.32 ^b^
APX (U/g FW)	Root	0.17 ± 0.02 ^b^	0.73 ± 0.04 ^a^	0.23 ± 0.01 ^b^
	Stem	0.64 ± 0.02 ^a^	0.91 ± 0.08 ^a^	0.85 ± 0.08 ^a^
	Leaf	2.93 ± 0.26 ^b^	4.59 ± 0.30 ^a^	3.19 ± 0.02 ^b^
GR (U/g FW)	Root	0.07 ± 0.01 ^b^	0.16 ± 0.01 ^a^	0.10 ± 0.02 ^b^
	Stem	0.10 ± 0.00 ^b^	0.18 ± 0.02 ^a^	0.16 ± 0.03 ^b^
	Leaf	0.12 ± 0.00 ^b^	0.21 ± 0.01 ^a^	0.16 ± 0.01 ^b^
GST (U/g FW)	Root	0.01 ± 0.00 ^b^	0.04 ± 0.01 ^a^	0.03 ± 0.01 ^ab^
	Stem	0.04 ± 0.00 ^c^	0.33 ± 0.01 ^a^	0.21 ± 0.01 ^b^
	Leaf	0.09 ± 0.00 ^b^	0.26 ± 0.01 ^a^	0.25 ± 0.02 ^a^
GSH (μmol/g FW)	Root	106.28 ± 9.40 ^b^	151.24 ± 3.58 ^a^	146.59 ± 5.52 ^a^
	Stem	190.91 ± 8.02 ^b^	357.58 ± 8.02 ^a^	321.21 ± 19.48 ^a^
	Leaf	785.86 ± 34.34 ^c^	1244.44 ± 69.33 ^a^	949.49 ± 8.81 ^b^
GSSG (μmol/g FW)	Root	46.46 ± 2.67 ^b^	92.02 ± 7.07 ^a^	70.71 ± 3.77 ^c^
	Stem	111.71 ± 4.10 ^c^	152.02 ± 2.61 ^a^	130.31 ± 3.72 ^b^
	Leaf	89.22 ± 4.32 ^b^	153.57 ± 7.87 ^a^	140.39 ± 5.19 ^a^
T-AOC (U/mL)	Root	126.10 ± 7.11 ^b^	363.62 ± 12.82 ^a^	323.45 ± 10.98 ^a^
	Stem	45.34 ± 2.47 ^c^	348.81 ± 13.62 ^b^	137.24 ± 6.57 ^b^
	Leaf	17.10 ± 3.91 ^b^	40.86 ± 6.34 ^a^	36.37 ± 2.72 ^ab^
MDA (nmol/L)	Root	6.10 ± 0.34 ^a^	3.76 ± 0.08 ^b^	5.47 ± 0.18 ^a^
	Stem	3.69 ± 0.12 ^a^	1.24 ± 0.2 ^c^	2.30 ± 0.08 ^b^
	Leaf	2.78 ± 0.19 ^a^	1.56 ± 0.12 ^b^	2.45 ± 0.25 ^a^
PRO (μg/g FW)	Root	17.01 ± 0.12 ^a^	4.54 ± 0.13 ^c^	6.80 ± 0.40 ^b^
	Stem	756.96 ± 22.87 ^a^	354.88 ± 69.18 ^b^	481.86 ± 55.12 ^b^
	Leaf	1960.32 ± 128.02 ^a^	714.29 ± 52.29 ^b^	718.82 ± 75.45 ^b^

Note: SOD: superoxide dismutase; APX: ascorbate peroxidase; GR: glutathione reductase; GST: glutathione S-transferase; GSH: reduced glutathione; GSSG: glutathiol; T-AOC: total antioxidant capacity; MDA: malondialdehyde; PRO: proline. Different lowercase letters represent significant differences within columns at the 0.05 level based on Scheffe’s test.

**Table 3 plants-12-01982-t003:** Statistics showing the number of differentially expressed genes.

DEG Set	No. of DEGs	Upregulated	Downregulated
2.5 Zn vs. 0 Zn	7291	3849	3442
40 Zn vs. 0 Zn	7781	3137	4644

## Data Availability

The data presented in this study are available on request from the corresponding author.

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
