# Peer review of "Spraying Zinc Sulfate to Reveal the Mechanism through the Glutathione Metabolic Pathway Regulates the Cadmium Tolerance of Seashore Paspalum (Paspalum vaginatum Swartz)"

_plants, 2023, doi:10.3390/plants12101982_

Round 1
Reviewer 1 Report
The article "Spraying zinc sulfate to reveal the mechanism through which the glutathione metabolic pathway regulates the cadmium tolerance of seashore paspalum" is well written and has novelty, which gathers a broad audience. A few changes are required:
1) Add some values of the studied attributes in the abstract.
2) Introduction is limited, try to add some more recent references in this section.
3) The objective of the study is hard to identify in the introduction. Please explain more about the objective of the study.
4) Figure 1 and 2 (Excel graphs) needs to be re-formulated. They don't look attractive at all. Change the font of these graphs.
5) Discussion is fragmented, try to make a story between genes regulations, enzymes, and physiology.
6) At what stage did you measure the photosynthetic attributes and which instruments were used, what time of the day? Please must indicate that information.
English needs to be improved some grammatical mistakes were observed.
Author Response
Reviewer(s)' Comments to Author:
# Reviewer 1
The article "Spraying zinc sulfate to reveal the mechanism through which the glutathione metabolic pathway regulates the cadmium tolerance of seashore paspalum" is well written and has novelty, which gathers a broad audience. A few changes are required:
1) Add some values of the studied attributes in the abstract.
Response:
We have made modifications as suggested.
- Introduction is limited, try to add some more recent references in this section.
Response:
We have made modifications as suggested, as shown in lines 63–68.
- The objective of the study is hard to identify in the introduction. Please explain more about the objective of the study.
Response:
We have made modifications as suggested (in lines 79–96). The objective of the present study is as follows: “To further improve the Cd resistance of seashore paspalum, we adopted the method of spraying ZnSO4. As a simple and easy way to improve the Cd tolerance of plants, ZnSO4 spray has been widely used. However, relevant studies only revealed the protective mechanism from a physiological perspective, and the tolerance mechanism of plants was not revealed. Therefore, we used physiological and transcriptome sequencing methods to reveal the mechanism controlling the Cd tolerance of seashore paspalum by spraying different concentrations of zinc sulfate.”
- Figure 1 and 2 (Excel graphs) needs to be re-formulated. They don't look attractive at all. Change the font of these graphs.
Response:
Figures 1 and 2 have been remade according to the suggestions.
5) Discussion is fragmented, try to make a story between genes regulations, enzymes, and physiology.
Response:
We have made modifications as suggested, as shown in lines 605–653.
- At what stage did you measure the photosynthetic attributes and which instruments were used, what time of the day? Please must indicate that information.
Response:
The relevant content has been added in lines679–684.

Reviewer 2 Report
First of all, we want to thank the authors fr this nice paper
The introduction need to be developed especialy on the foliar application of zinc sulfate and why u use it.
why u chosse these 2 concentration - the differance between them is very high 0.5 then 40 - strange choice.
The bioinformatic work is very good.
symmary disscussion figure is ok and clear, but u have to give recommendation in the end to spray 0.5 or 40 ????
Why the net photosynthesis is decreased at Zn 40 after being increased in Zn 0.5.
Also the same in the antioxident - go up in low con. then go down in high Zn
The quality of the English Language is ok, only some tempo erros
Author Response
# Reviewer 2
First of all, we want to thank the authors fr this nice paper
- The introduction need to be developed especially on the foliar application of zinc sulfate and why u use it.
Response:
We have made modifications as suggested, as shown in lines 83–96. The reason for the foliar application of zinc sulfate is as follows: “Studies have shown that the plant toxicity of Cd can be inhibited by the interaction of cations (such as Zn2+) during the root uptake of Cd. In traditional agricultural practices, zinc sulfate (ZnSO4) or zinc chelate of ethylenediamine tetraacetic acid chelate (ZnEDTA) is applied to leaves and the ground, while foliar zinc application is an effective method to promote plant absorption of zinc. A study on Triticum aestivum cv. Shield plants showed that the proportion of zinc on leaves treated with ZnSO4 was significantly higher than that of leaves treated with ZnEDTA. To further improve the Cd resistance of seashore paspalum, we adopted the method of spraying ZnSO4. As a simple and easy way to improve the Cd tolerance of plants, ZnSO4 spray has been widely used. However, relevant studies only revealed the protective mechanism from a physiological perspective, and the tolerance mechanism of plants was not revealed. Therefore, we used physiological and transcriptome sequencing methods to reveal the mechanism controlling the Cd tolerance of seashore paspalum by spraying different concentrations of zinc sulfate.”
- why u chosse these 2 concentration - the differance between them is very high 0.5 then 40 - strange choice.
Response:
We conducted preliminary experiments and set a series of concentration gradients (0, 2.5, 5, 10, 20, and 40 mM ZnSO4). We found that among all the sprayed zinc sulfate solutions, plants treated with 2.5 mM ZnSO4 showed the strongest resistance under cadmium stress, while plants treated with 40 mM ZnSO4 showed the weakest resistance. Therefore, we conducted research on these two concentrations.
3) The bioinformatic work is very good.
symmary disscussion figure is ok and clear, but u have to give recommendation in the end to spray 0.5 or 40 ????
Response:
Relatively speaking, compared to 40 mM ZnSO4, the cadmium content in different parts of the 0.5 mM ZnSO4 plant was remarkably reduced, showing strong antioxidant capacity. We suggest spraying 0.5 mM ZnSO4 to improve the cadmium tolerance of seashore paspalum.
4) Why the net photosynthesis is decreased at Zn 40 after being increased in Zn 0.5.
Also the same in the antioxident - go up in low con. then go down in high Zn.
Response:
Compared to low concentration treatment (2.5 mM), plants under high concentration ZnSO4 treatment (40 mM) may have been exposed to zinc toxicity, resulting in a decrease in their photosynthetic and antioxidant abilities.

Reviewer 3 Report
The paper “Spraying zinc sulfate to reveal the mechanism through which the glutathione metabolic pathway regulates the cadmium tolerance of seashore paspalum (Paspalum vaginatum Swartz)” by Cui Liwen, Chen Yu, Lliu Jun, Zhang Qiang, Xu Lei and Yang Zhimin describes the study of seashore paspalum behaviour under two levels of cadmium stress (plus control) followed by gene expression study and revealing of the protection mechanism of zinc sulfate against Cd poisoning.
Selected model plant is known to handle salinity stress very well and, apparently, can survive rather contaminated soil conditions. Zn and Cd are very close from the viewpoint of chemical behaviour, and they compete in the in vivo system, similarly to other pairs of elements, like Na + K or Mg + Ca. However, while Cd is undesirable, toxic pollutant, Zn is a micronutrient and its concentration must be kept within certain range for optimal functioning of the plant.
Understanding the effect of Zn to prospective Cd uptake and mechanisms that regulate such a process in of utmost importance, especially in the world still burdened by residua from NiCd batteries, photographic solutions a similar sources used widespread just a few decades ago.
The paper is given in good, simple scientific English. The experiment is interesting and it brought important results, it is described clearly, with appropriate amount of tables and figures in very good quality. It will be beneficial for the readers of Plants, and should be accepted.
The manuscript needs a little technical editing. Lines 159-172 are empty in *.pdf. Lines 255-278 will have to be hand-hadled, because automatic justification was not able to divide the words well. Lines 421 – 448 are empty, followed by a hard enter. These bugs are caused by incompatibility among PC systems and text processors and it is not authors’ fault, but let them fix it. They will find more cases, like line 610, themselves. Everybody wants their paper look good.
The introduction is rather comprehensive, but it only describes one side of the coin. The uptake of Cd by plants is a problem that should be reduced. However, it would be great to remove the Cd contamination from agricultural-used soil, and there are studies of heavy metal uptake by plants that can be used for “phytoremediation”. Some types of vegetables are particularly good with cadmium. Is seashore paspalum competitive? Are there some phytoremediation uses? You might want to mention it in the text.
Author Response
# Reviewer 3
The paper “Spraying zinc sulfate to reveal the mechanism through which the glutathione metabolic pathway regulates the cadmium tolerance of seashore paspalum (Paspalum vaginatum Swartz)” by Cui Liwen, Chen Yu, Lliu Jun, Zhang Qiang, Xu Lei and Yang Zhimin describes the study of seashore paspalum behaviour under two levels of cadmium stress (plus control) followed by gene expression study and revealing of the protection mechanism of zinc sulfate against Cd poisoning.
Selected model plant is known to handle salinity stress very well and, apparently, can survive rather contaminated soil conditions. Zn and Cd are very close from the viewpoint of chemical behaviour, and they compete in the in vivo system, similarly to other pairs of elements, like Na + K or Mg + Ca. However, while Cd is undesirable, toxic pollutant, Zn is a micronutrient and its concentration must be kept within certain range for optimal functioning of the plant.
Understanding the effect of Zn to prospective Cd uptake and mechanisms that regulate such a process in of utmost importance, especially in the world still burdened by residua from NiCd batteries, photographic solutions a similar sources used widespread just a few decades ago.
The paper is given in good, simple scientific English. The experiment is interesting and it brought important results, it is described clearly, with appropriate amount of tables and figures in very good quality. It will be beneficial for the readers of Plants, and should be accepted.
The manuscript needs a little technical editing. Lines 159-172 are empty in *.pdf. Lines 255-278 will have to be hand-hadled, because automatic justification was not able to divide the words well. Lines 421 – 448 are empty, followed by a hard enter. These bugs are caused by incompatibility among PC systems and text processors and it is not authors’ fault, but let them fix it. They will find more cases, like line 610, themselves. Everybody wants their paper look good.
The introduction is rather comprehensive, but it only describes one side of the coin. The uptake of Cd by plants is a problem that should be reduced. However, it would be great to remove the Cd contamination from agricultural-used soil, and there are studies of heavy metal uptake by plants that can be used for “phytoremediation”. Some types of vegetables are particularly good with cadmium. Is seashore paspalum competitive? Are there some phytoremediation uses? You might want to mention it in the text.
Response:
Because they are the primary source of human food, researchers have tried to reduce the accumulation of cadmium in the plants of vegetable crops. Because lawn grass plants are not readily edible , they are more commonly used in environmental protection. The purpose of this study was to investigate the mechanism through which zinc sulfate spray promotes cadmium tolerance in seashore paspalum regardless of the soil remediation ability. The soil remediation ability of seashore paspalum has been studied in our other experiments. In another of our studies, we found a significant increase in the cadmium enrichment coefficient after applying nitrate nitrogen fertilizer. In our future work, we will attempt to further analyze the enrichment ability of vegetable crops.

Round 2
Reviewer 2 Report
The paper become much better
Congratulation
It is ok